# Optimal Convergence Rates of Deep Convolutional Neural Networks: Additive Ridge Functions

**Zhiying Fang**                                                              *fangzhiying@szpt.edu.cn*
*Institute of Applied Mathematics*
*Shenzhen Polytechnic*[1]
*School of Data Science*
*The Chinese University of Hong Kong, Shenzhen*[2]

**Guang Cheng**                                                              *guangcheng@stat.ucla.edu*
*Department of Statistics*
*University of California, Los Angeles*

**Reviewed on OpenReview:** *https://openreview.net/forum?id=Q6ZXm7VBFY*

## Abstract

Convolutional neural networks have shown impressive abilities in many applications, especially those related to the classification tasks. However, for the regression problem, the abilities of convolutional structures have not been fully understood, and further investigation is needed. In this paper, we consider the mean squared error analysis for deep convolutional neural networks. We show that, for additive ridge functions, convolutional neural networks followed by one fully connected layer with ReLU activation functions can reach optimal mini-max rates (up to a log factor). The input dimension only appears in the constant of convergence rates. This work shows the statistical optimality of convolutional neural networks and may shed light on why convolutional neural networks are able to behave well for high dimensional input.

## 1 Introduction

Deep learning, based on deep neural networks structures and elaborate optimization techniques, has achieved great successes and empirically outperformed classical machine learning methods such as kernel methods in many applications in areas of science and technology (Kingma & Ba, 2014; Le et al., 2011; LeCun et al., 2015; Schmidhuber, 2015). Especially, convolutional neural networks were considered to be the most powerful tool for various tasks such as image classification (Goodfellow et al., 2016), speech recognition (Hinton et al., 2006) and sequence analysis in bio-informatics (Alipanahi et al., 2015; Zhou & Troyanskaya, 2015) before the appearance of the Transformer structures (Vaswani et al., 2017; Dosovitskiy et al., 2020). However, compared with significant achievements and developments in practical applications, theoretical insurances are left behind.

Recently, many researchers in various fields have done great works trying to explain the mysteries of convolutional neural networks (d'Ascoli et al., 2019; Neyshabur, 2020), and most of the existing works focus on the approximation properties of convolutional structures. For vector inputs, a universality result for convolutional neural networks is first established in Zhou (2020b). Afterwards, Zhou (2020a) further illustrates that deep convolutional neural networks are at least as good as fully connected neural networks in the sense that any output of a fully connected neural network can be reconstructed by a deep convolutional neural network with the same order of free parameters. It is also shown in Fang et al. (2020) that functions in Sobolev spaces on the unit sphere or taking an additive ridge form can be approximated by deep convolutional neural networks with optimal rates. For square matrix inputs, authors in Petersen & Voigtlaender (2020) prove

that if the target function is translation equivariant, then convolutional structures are equivalent to fully connected ones in terms of approximation rates with periodic convolution. Recently, He et al. (2021) derives a decomposition theorem for large convolutional kernels and enables convolutional networks to duplicate one hidden layer neural networks by deep convolutional nets including structures of ResNet and MgNet.

For approximation ability of fully connected neural networks, Yarotsky (2017) establishes upper and lower bounds of the complexity of deep ReLU networks when approximating functions in Sobolev spaces. Petersen & Voigtlaender (2018) also derives optimal approximation ability for piecewise smooth functions with fixed layer ReLU networks. For functions in Besov spaces, approximation results are presented in Suzuki (2018). Compared with numerous works concerning mean squared error analysis of fully connected neural networks (Suzuki, 2018; Imaizumi & Fukumizu, 2019; Schmidt-Hieber et al., 2020), only very few papers attempt to understand the convolutional structures from a statistical learning perspective. With the same spirit in Zhou (2020a), authors in Oono & Suzuki (2019) show that any block-sparse fully connected neural network with $M$ blocks can be realized by a ResNet-type convolutional neural network with fixed-sized channels and filters by adding $O(M)$ parameters. Consequently, if a function class can be approximated by block sparse fully connected nets with optimal rates, then it can also be achieved by ResNet-type convolutional nets. Authors prove that ResNet-type convolutional neural networks can reach optimal convergence rates for functions in Barron and Hölder classes by first transforming a fully connected net to a block sparse type, then to ResNet-type convolutional nets. Authors in Mao et al. (2021) first establish approximation results for deep convolutional neural networks with one fully connected layer when the target function takes a composite form as $f \circ Q$ with a feature polynomial $Q$ and a univariate function $f$. Two groups of convolutional structures are applied to approximate functions $Q$ and $f$ consecutively. Then mean squared error rates are derived for the function class $f \circ Q$. Analysis shows that the estimation error decreases to a minimum as the network depth increases to an optimal value, and then increases as the depth becomes larger. According to the minimax lower bound derived in our paper, the rate is sub-optimal. Universal consistency of pure convolutional structures is recently presented by Lin et al. (2021) in the framework of empirical risk minimization.

Compared with previous works on mean squared error analysis of deep convolutional neural networks, we show that functions possessing inherent structures $\xi \cdot x$ in an additive form can be directly learned by deep convolutional networks. Formally, we consider mean squared error analysis for deep ReLU convolutional neural networks in the estimation of additive ridge functions. The function class takes the form of

$$\sum_{i=1}^{m} f_i(\xi_i \cdot x),$$

where for each $i$, $f_i$ satisfies some regularity conditions, $\xi_i$ can be considered as a projection vector and $\cdot$ is the inner product in $\mathbb{R}^d$. This function class is also known as the additive index models (Yuan, 2011) in statistics and related to the projection pursuit regression introduced by Friedman & Stuetzle (1981). It has been shown in Diaconis & Shahshahani (1984) that this function class can be used to approximate any square-integrable function to arbitrary precision. A precise definition will be presented in Section 3.

For additive ridge functions, various investigations have been done including convergence rates, identifiability, iterative estimation in Chen (1991); Chiou & Müller (2004); Ruan & Yuan (2010); Yuan (2011); Bach (2017). Also, with a similar structure to the function class above, minimax convergence rates have been presented in Klusowski & Barron (2016; 2017) when all the $f_i$ are sine functions or Hermite polynomials. However, none of these works presents mini-max lower rate for this type of functions when $f_i$ possess different regularities. Though optimal convergence rates are achieved in Ruan & Yuan (2010) by traditional reproducing kernels with regularized least squares scheme, shortcomings of this method are quite obvious when comparing to neural networks. For example, to achieve optimal convergence rates, the best regularization parameter is usually related to the regularity of the target function which is practically unknown and is often decided by cross validation. However, convolutional neural networks are more automatic in the sense that the function smoothness is not required when implementing the algorithm. Also, the smoothing spline algorithm is applied in a backfitting manner, whereas the convolutional neural network is more generic.

We show that deep convolutional neural networks followed by one fully connected layer are able to achieve the mini-max optimal convergence rates for additive ridge functions by a careful analysis of the covering number of deep convolutional structures. The additive index $m$ would affect the depth of our networks in a linear

way which indicates the importance of depth of neural networks when the target function is complicated. The inherent structures $\xi_i \cdot x$ can represent different localized features of input $x$ when $\xi_i$s are sparse. The input dimension only appears in the constant of the final convergence rate.

In summary, the contributions of our work are as follows:

- We conduct an in-depth covering number analysis for deep convolutional neural networks presented in Fang et al. (2020). Thanks to the simple structure and few free parameters of convolutional neural networks, we derive small complexity of this hypothesis space.

- We present a mini-max lower bound for additive ridge functions. As a direct consequence, the lower bound also applies when $\xi \cdot x$ is generalized to any polynomial function $Q(x)$. We show that for these two types of functions, lower rates are dimension independent.

- By combining the approximation result in Fang et al. (2020) and our covering number analysis, we show that deep ReLU convolutional neural networks followed by one fully connected layer can reach optimal convergence rates for the regression problem for additive ridge functions and the input dimension only appears in the constant of the upper bound.

## 2 Problem setting

We first give a brief introduction to the structure of convolutional neural networks considered in this work.

For a sequence $w = (w_k)_{k\in\mathbb{Z}}$ supported in $\{0, 1, \cdots, S\}$ and another one $x = (x_k)_{k\in\mathbb{Z}}$ supported in $\{1, 2, \cdots, D\}$, the convolution of these two sequences is given by

$$(w * x)_i = \sum_{k\in\mathbb{Z}} w_{i-k}x_k = \sum_{k=1}^{D} w_{i-k}x_k, \qquad i \in \mathbb{Z},$$

which is supported in $\{1, \cdots, D + S\}$.

By applying this convolutional operation, we know that the matrix in each layer of convolutional neural networks should be of the form

$$T^w = \begin{bmatrix} w_0 & 0 & 0 & 0 & \cdots & \cdots & 0 \\ w_1 & w_0 & 0 & 0 & \cdots & \cdots & 0 \\ \vdots & \ddots & \ddots & \ddots & \ddots & \ddots & \vdots \\ w_S & w_{S-1} & \cdots & w_0 & 0 & \cdots & 0 \\ 0 & w_S & \cdots & w_1 & w_0 & 0\cdots & 0 \\ \vdots & \ddots & \ddots & \ddots & \ddots & \ddots & \vdots \\ 0 & \cdots & 0 & w_S & \cdots & w_1 & w_0 \\ 0 & \cdots & 0 & 0 & w_S & \cdots & w_1 \\ \vdots & \ddots & \ddots & \ddots & \ddots & \ddots & \vdots \\ 0 & 0 & 0 & 0 & \cdots & 0 & w_S \end{bmatrix}, \tag{1}$$

with $T^w \in \mathbb{R}^{(D+S)\times D}$. For input $x = (x_1, x_2, \cdots, x_d) \in \mathbb{R}^d$, a deep convolutional neural network with $J$ hidden layers $\{h^{(j)} : \mathbb{R}^d \to \mathbb{R}^{d_j}\}$ and widths $\{d_j = d_{j-1} + S^{(j)}\}$ can be defined iteratively by $h^{(0)}(x) = x$ and

$$h^{(j)}(x) = \sigma\left(T^{(j)} h^{(j-1)}(x) - b^{(j)}\right), \qquad j = 1, \ldots, J,$$

if we denote $T^{(j)} := T^{w^{(j)}}$ with $D = d_{j-1}$ and $S = S^{(j)}$ for $j = 1, \cdots, J$. Throughout this paper, we take identical filter length $S^{(j)} = S \in \mathbb{N}$ which implies $\{d_j = d + jS\}$ and take the activation function as the ReLU,

$$\sigma(x) = \max\{0, x\}, \qquad x \in \mathbb{R}.$$

Since the sums of the rows in the middle of the Toeplitz type matrix (1) are equal, we impose a restriction for the bias vectors $\{b^{(j)}\}_{j=1}^{J}$ of the convolutional layers

$$b^{(j)}_{S+1} = \ldots = b^{(j)}_{d_j-S}, \qquad j = 1, \ldots, J. \tag{2}$$

After the last convolutional layer, we add one fully connected layer $h^{(J+1)}$ with a restricted matrix $F^{(J+1)}$ and a bias vector $b^{(J+1)}$. Precisely, we have

$$h^{(J+1)}(x) = \sigma\left(F^{(J+1)}h^{(J)}(x) - b^{(J+1)}\right).$$

For the fully connected layer, we take $F^{(J+1)}_{(j-1)(2N+3)+i} = 1$ for $j = 1, \ldots, m$, $i = 1, \ldots, 2N+3$ and 0 otherwise. We let $\mathbf{1}_{2N+3} = (1, 1, \ldots, 1)^T \in \mathbb{R}^{2N+3}$. Specifically, the matrix in the last layer takes the form of

$$F^{(J+1)} = \begin{bmatrix} O & \mathbf{1}_{2N+3} & O & O_{2N+3} & \cdots & O & O_{2N+3} & O \\ O & O_{2N+3} & O & \mathbf{1}_{2N+3} & \cdots & O & O_{2N+3} & O \\ \vdots & \vdots & \vdots & \ddots & \ddots & \ddots & \ddots & \vdots \\ O & O_{2N+3} & O & O_{2N+3} & \cdots & O & \mathbf{1}_{2N+3} & O \end{bmatrix}, \tag{3}$$

where $F^{(J+1)} \in \mathbb{R}^{(2N+3)m \times (d+JS)}$ for some positive integer $N \in \mathbb{N}$ which can be considered as the order of the number of free parameters in the network.

We restrict the full matrix to take a simple form as (3) to further demonstrate abilities of convolutional structures. The hypothesis space induced by our network is given by

$$\mathcal{H} := \mathcal{H}_{J,B,S,N} := \left\{c \cdot h^{(J+1)}(x) : \left\|w^{(j)}\right\|_{\infty} \le B, \left\|b^{(j)}\right\|_{\infty} \le 2((S+1)B)^j, \|c\|_{\infty} \le NB\right\}, \tag{4}$$

and $F^{(J+1)}$ takes the form (3) with $\left\|F^{(J+1)}\right\|_{\infty} \le 1$. Here $B$ is a constant depending on the target function space and the filter size $S$ that will be given explicitly in **Lemma 3.4**.

## 2.1 Statistical learning framework

Now we formulate regression problems in the setting of statistical learning theory.

Let $\mathcal{X}$ be the unit ball in $\mathbb{R}^d$, that is, $\mathcal{X} := \{x : \|x\|_2 \le 1, x \in \mathbb{R}^d\}$ and $\mathcal{Y} \subset \mathbb{R}$. In the non-parametric regression model, we observe $n$ i.i.d. vectors $x_i \in \mathcal{X}$ and $n$ responses $y_i \in \mathbb{R}$ from the model

$$y_i = f^*(x_i) + \epsilon_i, \quad i = 1, \cdots, n,$$

where the noise variables $\epsilon_i$ are assumed to satisfy $\mathbb{E}(\epsilon_i|x_i) = 0$ and it is common to assume standard normal distribution for the noise variables. Our goal is to recover the function $f^*$ from the sample $\{(x_i, y_i)\}_{i=1}^{n}$.

In statistical learning theory, the regression framework is described by some unknown probability measure $\rho$ on the set $\mathcal{Z} = \mathcal{X} \times \mathcal{Y}$. The target is to learn the regression function $f_\rho(x)$ given by

$$f_\rho(x) = \int_{\mathcal{Y}} y d\rho(y|x), \quad x \in \mathcal{X},$$

where $\rho(y|x)$ denotes the conditional distribution of $y$ at point $x$ induced by $\rho$. It is easy to see that these two settings are equivalent by letting

$$f_\rho(x) = f^*(x).$$

For any measurable function, we define the $L^2$ prediction error as

$$\mathcal{E}(f) := \int_{\mathcal{Z}} (f(x) - y)^2 d\rho,$$

and clearly we know that $f_\rho(x)$ is the minimizer of it among all measurable functions. We assume that the sample $D = \{(x_i, y_i)\}_{i=1}^n$ is drawn from the unknown probability measure $\rho$. Let $\rho_\mathcal{X}$ be the marginal distribution of $\rho$ on $\mathcal{X}$ and $\left(L_{\rho_\mathcal{X}}^2, \|\cdot\|_{\rho_\mathcal{X}}\right)$ be the space of $\rho_\mathcal{X}$ square-integrable functions on $\mathcal{X}$. For any $f \in L_{\rho_\mathcal{X}}^2$, by a simple calculation we know that

$$\mathcal{E}(f) - \mathcal{E}(f_\rho) = \|f - f_\rho\|_{\rho_\mathcal{X}}^2 .$$

Since the distribution $\rho$ is unknown, we can not find $f_\rho$ directly. We thus use

$$f_{D,\mathcal{H}} = \arg\min_{f \in \mathcal{H}} \mathcal{E}_D(f), \tag{5}$$

to approximate $f_\rho$ where $\mathcal{H}$ is our hypothesis space and $\mathcal{E}_D(f)$ is defined to be the empirical risk induced by the sample given by

$$\mathcal{E}_D(f) := \frac{1}{n} \sum_{i=1}^n (f(x_i) - y_i)^2 .$$

The aim of mean squared error analysis is to derive convergence rate of $\|f_{D,\mathcal{H}} - f_\rho\|_{\rho_\mathcal{X}}^2$. We assume that $|y| \leq M$ almost everywhere and we have $|f_\rho(x)| \leq M$. We project the output function $f : \mathcal{X} \to \mathbb{R}$ onto the interval $[-M, M]$ by a projection operator

$$\pi_M f(x) = \begin{cases} f(x), & \text{if } -M \leq f(x) \leq M, \\ M, & \text{if } f(x) > M, \\ -M, & \text{if } f(x) < M, \end{cases}$$

and we consider $\pi_M f_{D,\mathcal{H}}$ as our estimator to $f_\rho(x)$. This type of clipping operator has been widely used in various statistical learning papers such as Suzuki (2018); Oono & Suzuki (2019); Mao et al. (2021).

## 2.2 Additive ridge functions

Additive ridge functions take the following form as

$$f(x) = \sum_{j=1}^m g_j(\xi_j \cdot x), \tag{6}$$

where $m$ is considered as a fixed constant. For each $j$, $g_j(\cdot) : \mathbb{R} \to \mathbb{R}$ is a univariate function and $\xi_j \cdot x$ represents the inner product in $\mathbb{R}^d$ with $\xi_j \in \mathbb{R}^d$ and $\|\xi_j\|$ is bounded by some constant $\Xi$. Since this constant $\Xi$ would only affect our results in a linear way, for simplicity of the proof, we take $\Xi = 1$. Specifically, we require that $0 < \|\xi_j\| \leq 1$ and we take $g_j \in W_\infty^\alpha ([-1, 1])$, the space of Lipschitz-$\alpha$ functions on $[-1, 1]$ with the semi-norm $|\cdot|_{W_\infty^\alpha}$ being the Lipschitz constant. We let $G = \max_{j=1, \cdots, m} \|g_j\|_\infty$. As mentioned before, (6) is also known as additive index models (Yuan, 2011) and related to the the projection pursuit regression (Friedman & Stuetzle, 1981). In particular, when $m = d$ and $(\xi_1, \cdots, \xi_m)$ is a permutation matrix, this function class reduces to the additive model (Hastie & Tibshirani, 2017) and it reduces to single index model (Duan & Li, 1991; Hardle et al., 1993; Ichimura, 1993) when $m = 1$. More precisely, we consider the function space

$$\begin{aligned} \Theta :=& \Theta_{m,\alpha,G,L} \\ :=& \{f(x) = \sum_{j=1}^m g_j(\xi_j \cdot x) : g_j \in W_\infty^\alpha ([-1, 1]), \\ & 0 < \|\xi_j\| \leq 1, \|g_j\|_\infty \leq G, \|g_j\|_{W_\infty^\alpha} \leq L\}, \end{aligned} \tag{7}$$

and we assume that the target function $f_\rho$ is in the set $\Theta$.

## 3 Main results

### 3.1 Covering number analysis of deep convolutional neural networks

The mean squared error analysis relies on the approximation abilities and the covering number of the hypothesis space. Before presenting the covering number analysis for the hypothesis space $\mathcal{H}$ (4), we would need to first state **Theorem 2** in Fang et al. (2020) which presents the approximation error to functions in the space (7) by deep convolutional neural networks.

**Theorem 3.1.** *Let $m \in \mathbb{N}$, $d \geq 3$, $2 \leq S \leq d$, $J = \left\lceil \frac{md-1}{S-1} \right\rceil$, and $N \in \mathbb{N}$. If $f \in \Theta$, then there exists a deep neural network consisting of $J$ layers of CNNs with filters of length $S$ and bias vectors satisfying (2) followed by one fully connected layer $h^{(J+1)}$ with width $m(2N+3)$ and connection matrix $F^{(J+1)}$ defined as (3) such that for some coefficient vector $c \in \mathbb{R}^{m(2N+3)}$ there holds*

$$\left\| f - c^{(J+1)} \cdot h^{(J+1)} \right\|_\infty \leq \sum_{j=1}^m |g_j|_{W_\infty^\alpha} N^{-\alpha}.$$

*The total number of free parameters $\mathcal{N}$ in the network can be bounded as*

$$\mathcal{N} \leq (3S+2) \left\lceil \frac{md-1}{S-1} \right\rceil + m(2N+2).$$

**Remark 3.2.** *In this convolutional neural network, each layer only contains one filter and padding is not considered. If we use multiple convolutional filters, then the target function can be more complicated. As a simple extension of the additive ridge functions, we can take the target function in the form of $\sum_{i=1}^m g_i(\sum_{j=1}^t \zeta_{i,j} \cdot x)$ if we consider $t$ filters in each layer. In this paper, the convolutional layers are used to learn the features $\xi_j \cdot x$ and the last fully connected layer is used to approximate functions $g_j$. Thus, the same approximation rate can also be achieved for this type of function by a two-layer fully connected networks with the same order of numbers of free parameters.*

To calculate the covering number of our target space, we first need Cauchy bound for polynomials and Vietas Formula to bound the infinity norm of filters in each layer. Proofs will be given in supplementary materials.

**Lemma 3.3.** *If $W = \{W_j\}_{j \in \mathbb{Z}}$ is a real sequence supported in $\{0, \cdots, K\}$ with $W_K = 1$, then all the complex roots of its symbol $\tilde{W}(z) = \sum_{j=0}^K W_j z^j$ are bounded by $1 + \max_{j=0,\cdots,K-1} |W_j|$, the Cauchy Bound of $\tilde{W}$. If we factorize $\tilde{W}$ into polynomials of degree at most $S$, then all the coefficients of these factor polynomials are bounded by $2^S (1 + \max_{j=0,\cdots,K-1} |W_j|)^S$.*

By applying the previous lemma, we are able to bound the magnitude of filters in each layer.

**Lemma 3.4.** *Let $2 \leq S \leq d$. For the deep convolutional neural networks constructed in this paper with $J$ convolutional layers and $1$ fully connected layer satisfying **Theorem 3.1**, there exists a constant $B = B_{\xi,S,G}$ depending on $\xi$, $S$ and $G$ such that $\|w^{(j)}\|_\infty \leq B$, $j = 1, \cdots, J_1$, and $\|F\|_\infty \leq 1$, $\|c^{(J+1)}\|_\infty \leq NB$, where $B$ is given by $B = \max\left\{ 2^S \left(1 + \left|\frac{1}{(\xi_m)_l}\right|\right)^S, 4G \right\}$.*

After bounding the filters, we can bound bias vectors and output functions in each layer.

**Lemma 3.5.** *Let $2 \leq S \leq d$. For the deep convolutional neural networks constructed in this paper with $J$ convolutional layers and $1$ fully connected layer satisfying **Theorem 3.1**, we have for $j = 1, \cdots, J+1$ that $\|b^{(j)}\|_\infty \leq 2\left((S+1)B\right)^j$, and*

$$\left\| h^{(j)}(x) \right\|_\infty \leq (2j+1)((S+1)B)^j. \tag{8}$$

After bounding all the filters, bias vectors and output functions in each layer, we can derive a bound for covering number of our hypothesis space $\mathcal{H}$ (4) as stated in the lemma below. The covering number $\mathcal{N}(\eta, \mathcal{H})$ of a subset $\mathcal{H}$ of $C(\mathcal{X})$ is defined for $\eta > 0$ to be the smallest integer $l$ such that $\mathcal{H}$ is contained in the union of $l$ balls in $C(\mathcal{X})$ of radius $\eta$. The notation of covering number can also be extended to $\mathcal{N}(\eta, \mathcal{H}, \star)$ where $\star$ denote some specific metric. $C(\mathcal{X})$ denotes the space of continuous functions on $\mathcal{X}$.

**Lemma 3.6.** *For $N \in \mathbb{N}$ and $\mathcal{H}$ given in (4), with two constants $C_{S,d,m,B}$ and $C''_{S,d,m,B}$ depending on $S, d, m, B$, there holds*

$$\log \mathcal{N}(\delta, \mathcal{H}) \leq C_{S,d,m,B} N \log \frac{1}{\delta} + C''_{S,d,m,B} N \log N,$$

*for any $0 < \delta \leq 1$.*

Constants $C_{S,d,m}$ and $C''_{S,d,m,B}$ depend on $m$, $S$ and $d$ at most in a cubic way. Since we treat $m$, $S$ and $d$ as fixed constants in our setting, the covering number is affected by $N$ in a linear way which is the order of free parameters in the last layer. The above lemma shows that the hypothesis space $\mathcal{H}$ has a relatively small covering number. Consequently, we are able to derive optimal convergence rates for convolutional neural networks.

### 3.2  Oracle inequality for empirical risk minimization

With the bound of the covering number of the hypothesis space, we are able to prove the oracle inequality which leads to the estimation error bound combining with approximation error. The following theorem presents the oracle inequality for empirical risk minimizers based on covering number estimates.

**Theorem 3.7.** *Suppose that $|y| \leq M$ almost everywhere and there exist constants $C_1, C_2 > 0$ and some real numbers $n_1, n_2 > 0$, such that*

$$\log \mathcal{N}(\delta, \mathcal{H}) \leq C_1 n_1 \log \frac{1}{\delta} + C_2 n_2 \log n_2, \quad \forall \delta > 0. \tag{9}$$

*Then for any $h \in \mathcal{H}$ and $\delta > 0$, we have*

$$\|\pi_M f_{D,\mathcal{H}} - f_\rho\|_{\rho_{\mathcal{X}}}^2 \leq \delta + 2 \|h - f_\rho\|_{\rho_{\mathcal{X}}}^2,$$

*holds with probability at least $1 - \exp\left\{C_1 n_1 \log \frac{16M}{\delta} - C_2 n_2 \log n_2 - \frac{3n\delta}{512M^2}\right\} - \exp\left\{\frac{-3n\delta^2}{16\left(3M + \|h\|_\infty\right)^2 \left(6\|h - f_\rho\|_{\rho_{\mathcal{X}}}^2 + \delta\right)}\right\}.$*

### 3.3  Mean squared error

By applying **Theorem 3.1** and **Theorem 3.7** we can obtain our main result on the upper bound of mean squared error.

**Theorem 3.8.** *Let $2 \leq S \leq d$, $0 < \alpha \leq 1$ and $\mathcal{H}, \Theta$ be defined as (4) and (7). If $|y| \leq M$ almost everywhere and $f_\rho \in \Theta$, then for $N \in \mathbb{N}$, we have*

$$\mathbb{E}\left\{\|\pi_M f_{D,\mathcal{H}} - f_\rho\|_{\rho_{\mathcal{X}}}^2\right\} \leq C \max\left\{N^{-2\alpha}, \frac{N \log N}{n}\right\},$$

*where the constant $C = C_{S,d,m,M,\alpha,B}$ is independent of the sample size $n$ and the $N$. In particular, if we choose $N = \left\lceil n^{\frac{1}{1+2\alpha}} \right\rceil$, then we can get*

$$\mathbb{E}\left\{\|\pi_M f_{D,\mathcal{H}} - f_\rho\|_{\rho_{\mathcal{X}}}^2\right\} \leq C n^{\frac{-2\alpha}{1+2\alpha}} \log n.$$

**Remark 3.9.** *The assumption of $|y| \leq M$ excludes the case of Gaussian noise and can be avoided by applying the oracle inequality Lemma 4 in Schmidt-Hieber et al. (2020). However, the above mentioned Lemma requires that the output functions of the neural network model are bounded by some constant $F$. This constant $F$ is usually related to the model complexity and thus may affect convergence rate if stated explicitly.*

*Proof of Theorem 3.8.* We know from **Theorem 3.1** that there exists some $h \in \mathcal{H}$ such that $\|h - f_\rho\|_{\rho_{\mathcal{X}}} \leq \|h - f_\rho\|_\infty \leq C_{\alpha,m} N^{-\alpha}$, where $C_{\alpha,m} = \sum_{j=1}^m |g_j|_{W_\infty^\alpha}$. We further know that $\|h\|_\infty \leq M + C_{\alpha,m}$. By applying **Theorem 3.7**, we have

$$\mathcal{E}(\pi_M f_{D,\mathcal{H}}) - \mathcal{E}(f_\rho) \leq 2 \|h - f_\rho\|_{\rho_{\mathcal{X}}}^2 + \epsilon,$$

holds with probability at least $1 - \exp\left\{C_1 N \log \frac{16M}{\epsilon} + C_2 N \log N - \frac{3n\epsilon}{512M^2}\right\} - \exp\left\{\frac{-3n\epsilon^2}{16(4M+C_{\alpha,m})^2\left(6C_{\alpha,m}^2 N^{-2\alpha}+\epsilon\right)}\right\}$ where $C_1 = C_{S,d,m,B}$ and $C_2 = C''_{S,d,m}$.

If we let

$$\epsilon \geq 6C_{\alpha,m}^2 N^{-2\alpha},$$

then we have

$$\mathcal{E}\left(\pi_M f_{D,\mathcal{H}}\right) - \mathcal{E}\left(f_\rho\right) \leq 2\epsilon,$$

hold with probability at least $1 - \exp\left\{C_3 N \log N - \frac{3n\epsilon}{512M^2}\right\} - \exp\left\{-\frac{3n\epsilon}{32(4M+C_{\alpha,m})^2}\right\}$, where $C_3 = (C_1 \log \frac{8M}{3C_{\alpha,m}^2} + 2\alpha C_1 + C_2)$.

We further let

$$\epsilon \geq \frac{1024 C_3 M^2 N \log N}{3n},$$

then we have

$$\mathcal{E}\left(\pi_M f_{D,\mathcal{H}}\right) - \mathcal{E}\left(f_\rho\right) \leq 2\epsilon,$$

holds with probability at least $1 - \exp\left\{-\frac{3n\epsilon}{1024M^2}\right\} - \exp\left\{-\frac{3n\epsilon}{32(4M+C_{\alpha,m})^2}\right\}$. By taking

$$C_4 = \max\left\{12C_{\alpha,m}^2, \frac{2048}{3}M^2 C_3, \frac{64}{3}(4M+C_{\alpha,m})^2\right\},$$

and $\tilde{\epsilon} = 2\epsilon$, we have

$$\mathbb{P}\left\{\mathcal{E}\left(\pi_M f_{D,\mathcal{H}}\right) - \mathcal{E}\left(f_\rho\right) \leq \tilde{\epsilon}\right\} \geq 1 - 2\exp\left\{-\frac{n\tilde{\epsilon}}{C_4}\right\},$$

for any $\tilde{\epsilon} \geq C_4 \max\left\{N^{-2\alpha}, \frac{N \log N}{n}\right\}$. Then by letting $\delta = 2\exp\left\{-\frac{n\tilde{\epsilon}}{C_4}\right\}$, we know that

$$\mathcal{E}\left(\pi_M f_{D,\mathcal{H}}\right) - \mathcal{E}\left(f_\rho\right) \leq C_9 \max\left\{\frac{N \log N}{n}, N^{-2\alpha}, \frac{\log\frac{2}{\delta}}{n}\right\},$$

holds with probability at least $1 - \delta$.

Now we apply $E\{\xi\} = \int_0^\infty \mathbb{P}\{\xi \geq t\} dt$ with $\xi = \mathcal{E}\left(\pi_M f_{D,\mathcal{H}}\right) - \mathcal{E}\left(f_\rho\right) = \|\pi_M f_{D,\mathcal{H}} - f_\rho\|_{\rho_\mathcal{X}}^2$. We have

$$\mathbb{E}\left[\|\pi_M f_{D,\mathcal{H}} - f_\rho\|_{\rho_\mathcal{X}}^2\right] \leq \int_0^T 1 dt + \int_T^\infty 2\exp\left\{\frac{-nt}{C_4}\right\} dt \leq T + \frac{2C_4}{n} \leq 3T,$$

with $T = C_4 \max\left\{N^{-2\alpha}, \frac{N \log N}{n}\right\}$. This finishes the proof with the constant $C_{S,d,m,M,\alpha,B}$ independent of $n$ or $N$ given by

$$C_{S,d,m,M,\alpha,B} = 3\max\left\{12C_{\alpha,m}^2, \frac{2048}{3}M^2 C_3, \frac{64}{3}(4M+C_{\alpha,m})^2\right\},$$

where $C_3 = (C_1 \log \frac{8M}{3C_{\alpha,m}^2} + 2\alpha C_1 + C_2)$, $C_1 = C''_{S,d,m,B}$ and $C_2 = C'_{S,d,m}$. $\qquad\square$

### 3.4 Lower bound for additive ridge functions

In this subsection, we will present the mini-max lower rate for estimating additive ridge functions. We let $\mathcal{M}(\rho, \Theta)$ be the class of all Borel measures $\rho$ on $\mathcal{X} \times \mathcal{Y}$ such that $f_\rho \in \Theta$. This class $\mathcal{M}(\rho, \Theta)$ is related to the set of distributions $\rho$ where data $\{\mathbf{x}_i, y_i\}_{i=1}^n$ is drawn from and $\Theta$ representing the set of target functions. Now we state our mini-max lower bound for the class $\mathcal{M}(\rho, \Theta)$.

**Theorem 3.10.** *Assume $m \geq 1$, $G > 0$ and $M \geq 4mG$. Let $\hat{f}_n(x)$ be the output of any learning algorithm based on the sample $\{\boldsymbol{x}_i, y_i\}_{i=1}^n$, then we have*

$$\inf_{\hat{f}_n} \sup_{\rho \in \mathcal{M}(\rho, \Theta)} \mathbb{E} \left\| \hat{f}_n(x) - f_\rho(x) \right\|_{L_{\rho_\mathcal{X}}^2}^2 \geq c_{m,G} n^{-\frac{2\alpha}{2\alpha+1}},$$

where the constant $c_{m,G}$ only depends on $m$ and $G$. For more discussion about this constant, please refer to the proof of **Theorem 3.9** in Appendix.

Since $\xi \cdot x$ is essentially a polynomial of $x$, we can obtain a mini-max lower bound for the function in the type of $f \circ Q$ as a direct consequence, where $Q(x)$ can be any polynomial of $x$. Formally, we define

$$\begin{aligned}
\Theta' :=& \Theta'(m, \alpha, G, L) \\
:=& \{f(x) = \sum_{j=1}^m g_j(Q(x)) : g_j \in W_\infty^\alpha([-1,1]), \\
& 0 < \|\xi_j\| \leq 1, \|g_j\|_\infty \leq G, \|g_j\|_{W_\infty^\alpha} \leq L\},
\end{aligned} \tag{10}$$

where $Q(x)$ denote any polynomial of $x$.

**Corollary 3.11.** *Assume $m \geq 1$, $G > 0$ and $M \geq 4mG$. Let $\hat{f}_n(x)$ be the output of any learning algorithm based on the sample $\{\boldsymbol{x}_i, y_i\}_{i=1}^n$, then we have*

$$\inf_{\hat{f}_n} \sup_{\rho \in \mathcal{M}(\rho, \Theta')} \mathbb{E} \left\| \hat{f}_n(x) - f_\rho(x) \right\|_{L_{\rho_\mathcal{X}}^2}^2 \geq c_{m,G} n^{-\frac{2\alpha}{2\alpha+1}},$$

*where the constant $c_{m,G}$ only depends on $m$ and $G$.*

Combining this lower bound with the upper bound in the last section, we know that deep convolutional neural networks followed by one fully connected layer can reach optimal convergence rates for additive ridge functions up to a log factor. In other words, for estimating $f_\rho \in \Theta$, no other methods could achieve a better rate than the estimator by deep convolutional neural networks. Furthermore, due to the special form of ridge functions, we can observe that this rate is dimension independent. Thanks to the simple structure of convolutional neural networks, we are able to derive small complexity bound and reach this rate with the input dimension appearing in the constant.

## 4 Conclusion and discussion

In this paper, we consider the regression problem in statistical learning theory by using deep convolutional neural networks. By a careful analysis of covering number of deep convolutional neural networks, we show that for additive ridge functions, deep convolutional neural networks followed by one fully connected layer can reach optimal convergence rates (up to a log factor). With the simple structure and few free parameters of convolutional neural networks, we obtain suitable complexity bound for the hypothesis space and are able to achieve this dimension independent convergence rate with the input dimension appearing in the constant, which shows the superiority of convolutional structure.

In the future we would like to address in what situation convolutional neural networks outperform usual fully connected neural networks concerning approximation abilities. Or can we derive optimal approximation rate for larger function classes when considering convolutional neural networks. Convolutional structures have been widely applied to various types of neural networks and have performed outstandingly in the classification problems. However, only few papers consider classification problems from a statistical perspective by using fully connected neural networks (Hu et al., 2020; Kim et al., 2021; Bos & Schmidt-Hieber, 2021). Also, there is currently no paper that directly approximates a specific function class (such as Sobolev space) through a two-dimensional convolutional neural network which is critical for the analysis of statistical learning theory. It would be significant if we can theoretically show the superiority of convolutional structures when considering classification problems with two dimensional convolution in the future.

## Acknowledgements

The authors would like to thank the referees for their encouraging comments and constructive suggestions. The first author is supported by the National Natural Science Foundation of China [NSFC-72150002]. The second author is supported by the Office of Naval Research [ONR N00014-22-1-2680] and National Science Foundation [NSF–SCALE MoDL (2134209)].

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

# A    Appendix: Proof of Main Results

**Proof of Theorem 3.1**

*Proof of Theorem 3.1.* The proof of this theorem is a combination of proofs of Lemma 3 and Theorem 2 in Fang et al. (2020) with minor modification. For the completeness of our following results, we present the proof here.

For $m \in \mathbb{N}$ and $\{\xi_1, \ldots, \xi_m\}$ with $0 < \|\xi_j\|_2 \le 1$, we take $W$ to be a sequence supported in $\{0, \cdots, md-1\}$ given by $W_{(j-1)d+(d-i)} = (\xi_j)_i$ where $j \in \{1, \cdots, m\}$ and $i \in \{1, \cdots, d\}$. By Lemma A.2 with $\mathcal{M} = md - 1$, there exists a sequence of filters $\mathbf{w} = \left\{w^{(j)}\right\}_{j=1}^{J}$ supported in $\{0, \cdots, S\}$ with $J \ge \lceil \frac{\mathcal{M}}{S-1} \rceil$ satisfying the convolutional factorization $W = w^{(J)} * w^{(J-1)} * \cdots * w^{(2)} * w^{(1)}$. Here for $j = p+1, \ldots, J$, we have taken $w^{(j)}$ to be the delta sequence $\delta_0$ given by $(\delta_0)_0 = 1$ and $(\delta_0)_k = 0$ for $k \in \mathbb{Z} \setminus \{0\}$. By Lemma A.3, we know that

$$T^{(J)}T^{(J-1)} \cdots T^{(1)} = T^{(J,1)} = (W_{i-k})_{i=1,\ldots,d+JS, k=1,\ldots,d} \in \mathbb{R}^{(d+JS) \times d},$$

where $T^{(j)}$ is the Toeplitz matrix with filter $w^{(j)}$ for $j = 1, 2, \ldots, J$.

Now we construct bias vectors for convolutional layers. We denote $\|w\|_1 = \sum_{k=-\infty}^{\infty} |w_k|$. We take $b^{(1)} = -\left\|w^{(1)}\right\|_1 \mathbf{1}_{d_1}$ and

$$b^{(j)} = \left(\Pi_{p=1}^{j-1} \left\|w^{(p)}\right\|_1\right) T^{(j)} \mathbf{1}_{d_{j-1}} - \left(\Pi_{p=1}^{j} \left\|w^{(p)}\right\|_1\right) \mathbf{1}_{d_{j-1}+S}, \tag{11}$$

for $j = 2, \cdots, J$. The bias vectors satisfy $b_{S+1}^{(j)} = \ldots = b_{d_j-S}^{(j)}$. Observe that $\|x\|_\infty \le 1$ for $x \in \mathcal{X}$. Denote $\|h\|_\infty = \max\{\|h_j\|_\infty : j = 1, \ldots, q\}$ for a vector of functions $h : \mathcal{X} \to \mathbb{R}^q$. We know that for $h : \mathcal{X} \to \mathbb{R}^{d_{j-1}}$,

$$\left\|T^{(j)}h\right\|_\infty \le \left\|w^{(j)}\right\|_1 \|h\|_\infty.$$

Hence the components of $h^{(J)}(x)$ satisfy

$$\left(h^{(J)}(x)\right)_{kd} = \langle \xi_k, x \rangle + B^{(J)}, \qquad k = 1, \ldots, m,$$

where $B^{(J)} = \Pi_{p=1}^{J} \left\|w^{(p)}\right\|_1$.

For the last fully connected layer, we have

$$h^{(J+1)}(x) = \sigma(F^{(J+1)}h^{(J)}(x) - b^{(J+1)}),$$

with the full matrix $F^{(J+1)}$ stated in (3). The bias vector in the last layer is given by $b^{(J+1)}_{(j-1)(2N+3)+i} = B^{(J)} + t_i$ for $j = 1, \ldots, m$, $i = 1, \ldots, 2N+3$ where $\mathbf{t} := \{t_1 < \cdots < t_{2N+3}\}$ is given in Lemma A.1. Then the fully-connected layer $h^{(J+1)}(x) \in \mathbb{R}^{m(2N+3)}$ of the deep network is

$$h^{(J+1)}_{(j-1)(2N+3)+i} = \sigma\left(\langle \xi_j, \cdot \rangle - t_i\right), \quad \text{for } 1 \le j \le m,\ 1 \le i \le 2N+3. \tag{12}$$

We choose the coefficient vector $c \in \mathbb{R}^{m(2N+3)}$ by means of the linear operator $\mathcal{L}_N$ (17) as

$$\left\{(c)_{(j-1)(2N+3)+i}\right\}^{2N+3}_{i=1} = N\mathcal{L}_N\left(\{g_j(t_i)\}^{2N+2}_{i=2}\right), \qquad j = 1, \ldots, m.$$

Then by the identity (18), we have

$$\begin{aligned}
c \cdot h^{(J+1)}(x) &= N \sum_{j=1}^{m} \sum_{i=1}^{2N+3} (c)_{(j-1)(2N+3)+i}\, \sigma\left(\langle \xi_j, x \rangle - t_i\right) \\
&= \sum_{j=1}^{m} L_{\mathbf{t}}\left(g_j\right)\left(\langle \xi_j, x \rangle\right).
\end{aligned}$$

Combining this with the additive ridge form (6) of $f$ and Lemma A.1, we know that for $x \in \mathcal{X}$,

$$\left| f(x) - c \cdot h^{(J+1)}(x) \right| = \left| \sum_{j=1}^{m} g_j(\langle \xi_j, x \rangle) - \sum_{j=1}^{m} L_{\mathbf{t}}\left(g_j\right)\left(\langle \xi_j, x \rangle\right) \right|$$

$$\le \sum_{j=1}^{m} \|g_j - L_{\mathbf{t}}(g_j)\|_{C[-1,1]} \le \sum_{j=1}^{m} |g_j|_{W^\alpha_\infty} N^{-\alpha}.$$

Then the desired error bound is verified.

$\square$

**Proof of Lemma 3.4**

*Proof of Lemma 3.4.* Since $0 < \|\xi_j\|_2 \le 1$, there exists some $l \in \{1, \cdots, d\}$ such that $(\xi_m)_l \neq 0$ and $(\xi_m)_i = 0$ for any $i < l$. Then we know that the sequence $W$ constructed in the proof of Theorem 3.1 is supported in $\{0, \cdots, md - l\}$ with $W_{md-l} = (\xi_m)_l \neq 0$. Now we set a sequence $W' = \frac{1}{|(\xi_m)_l|} W$, then $W'$ satisfies the condition in Lemma 3.3 with $K = md - l$. By Lemma 3.3, all the complex roots of $W'$ are located in the disk of radius $1 + \max_{j=0,\cdots,md-1}\left|\frac{W_j}{(\xi_m)_l}\right| \le 1 + \left|\frac{1}{(\xi_m)_l}\right|$, and the filters $\{w^{(j)}\}^J_{j=1}$ constructed in our networks satisfying Lemma A.2 can be bounded as

$$\left\|w^{(j)}\right\|_\infty \le 2^S \left(1 + \left|\frac{1}{(\xi_m)_l}\right|\right)^S, \quad j = 1, \cdots, J.$$

For the matrix $F^{(J+1)}$, we have $\left\|F^{(J+1)}\right\|_\infty \le 1$. For the vector $c$ in the hypothesis space, we know that from the construction in the proof of Theorem 3.1 we have $\|c\|_\infty \le 4NG$. Then we can take $B$ as

$$B = \max\left\{2^S \left(1 + \left|\frac{1}{(\xi_m)_l}\right|\right)^S, 4G\right\}.$$

$\square$

**Proof of Lemma 3.5**

*Proof of Lemma 3.5.* The bias vectors $\left\{b^{(j)}\right\}_{j=1}^{J}$ of CNNs are chosen in the proof of Theorem 3.1 as $b^{(1)} = -\left\|w^{(1)}\right\|_{1} \mathbf{1}_{d_1}$ and clearly we have $\left\|b^{(1)}\right\|_{\infty} \leq (S+1)B \leq 2(S+1)B$. For $j = 2, \cdots, J$, we have $b^{(j)} = \left(\Pi_{p=1}^{j-1}\left\|w^{(p)}\right\|_{1}\right)T^{(j)}\mathbf{1}_{d_{j-1}} - \left(\Pi_{p=1}^{j}\left\|w^{(p)}\right\|_{1}\right)\mathbf{1}_{d_{j-1}+S}$. Thus, we know that

$$\left\|b^{(j)}\right\|_{\infty} \leq ((S+1)B)^{j-1}\left\|T^{(j)}\mathbf{1}_{d_{j-1}}\right\|_{\infty} + ((S+1)B)^{j} \leq 2((S+1)B)^{j}.$$

For $b^{(J+1)}$ constructed in Theorem 3.1, clearly we have $\left\|b^{(J+1)}\right\|_{\infty} \leq 2\left((S+1)B\right)^{J} + 2 \leq 2\left((S+1)B\right)^{J+1}$. If $w$, $b$ in each layer and $F$, $c$ satisfy the restrictions in (4), then by Lipschitz condition of ReLU and the special form of Toeplitz matrix $T$, we have

$$\left\|h^{(j)}(x)\right\|_{\infty} \leq (S+1)B\left\|h^{(j-1)}(x)\right\|_{\infty} + 2((S+1)B)^{j},$$

for $j = 1, \cdots, J$. Combining this with $\left\|h^{(0)}\right\|_{\infty} \leq 1$, we can obtain by induction

$$\left\|h^{(j)}(x)\right\|_{\infty} \leq (2j+1)((S+1)B)^{j}.$$

It can be checked easily that the above inequality holds when $j = J+1$ by the special form of $F^{(J+1)}$.

$\square$

**Proof of Lemma 3.6**

*Proof of Lemma 3.6.* If $\hat{c} \cdot \hat{h}^{(J+1)}(x)$ is another function from the hypothesis space $\mathcal{H}$ induced by $\hat{\mathbf{w}}, \hat{\mathbf{b}}, \hat{F}^{(J+1)}, \hat{c}$ satisfying the restrictions in (4) and

$$\left\|w^{(j)} - \hat{w}^{(j)}\right\|_{\infty} \leq \delta, \left\|b^{(j)} - \hat{b}^{(j)}\right\|_{\infty} \leq \delta,$$
$$\|c - \hat{c}\|_{\infty} \leq \delta, \left\|F^{(J+1)} - \hat{F}^{(J+1)}\right\|_{\infty} \leq \delta,$$

then by the Lipschitz property of ReLU, we have

$$\left\|h^{(j)} - \hat{h}^{(j)}\right\|_{\infty} \leq \left\|\left(T^{w^{(j)}}h^{(j-1)}(x) - b^{(j)}\right) - \left(T^{\hat{w}^{(j)}}\hat{h}^{(j-1)}(x) - \hat{b}^{(j)}\right)\right\|_{\infty}$$
$$\leq \left\|T^{w^{(j)}}\left(h^{(j-1)}(x) - \hat{h}^{(j-1)}(x)\right)\right\|_{\infty} + \left\|\left(T^{w^{(j)}} - T^{\hat{w}^{(j)}}\right)\hat{h}^{(j-1)}(x)\right\|_{\infty} + \delta,$$

for $j = 1, \cdots, J$. Combining the above equation with (8) and the special form of the Toeplitz matrix $T^{w^{(j)}} - T^{\hat{w}^{(j)}} = T^{w^{(j)} - \hat{w}^{(j)}}$, we know that

$$\left\|h^{(j)} - \hat{h}^{(j)}\right\|_{\infty} \leq (S+1)B\left\|h^{(j-1)} - \hat{h}^{(j-1)}\right\|_{\infty} + (S+1)\delta(2j-1)((S+1)B)^{j-1} + \delta.$$

Further by induction and $h^{(0)} - \hat{h}^{(0)} = 0$, we have

$$\left\|h^{(j)} - \hat{h}^{(j)}\right\|_{\infty} \leq (j^2 + j)((S+1)B)^{j}\delta,$$
$$\leq 2j^2((S+1)B)^{j}\delta, \quad j = 1, \cdots, J.$$

For the last fully connected layer and we know that

$$\left\|h^{(J+1)} - \hat{h}^{(J+1)}\right\|_\infty$$
$$\leq \left\|\left(F^{J+1}h^{(J)}(x) - b^{(J+1)}\right) - \left(\hat{F}^{J+1}\hat{h}^{(J)}(x) - \hat{b}^{(J+1)}\right)\right\|_\infty$$
$$\leq \left\|F^{J+1}\left(h^{(J)}(x) - \hat{h}^{(J)}(x)\right)\right\|_\infty + \left\|\left(F^{J+1} - \hat{F}^{J+1}\right)\hat{h}^J(x)\right\|_\infty + \delta$$
$$\leq 2J^2((S+1)B)^J\delta + \delta(2J+1)((S+1)B)^J + \delta$$
$$\leq 5J^2((S+1)B)^J\delta.$$

Finally for the output function, we have

$$\left\|c \cdot h^{(J+1)} - \hat{c} \cdot \hat{h}^{(J+1)}\right\|_\infty$$
$$\leq \left\|c \cdot \left(h^{(J+1)}(x) - \hat{h}^{(J+1)}(x)\right)\right\|_\infty + \left\|(c - \hat{c}) \cdot \hat{h}^{(J+1)}(x)\right\|_\infty$$
$$\leq (2N+3)mNB5J^2((S+1)B)^J\delta + (2N+3)m\delta(2J+3)((S+1)B)^{J+1}$$
$$\leq 6(J^2+4)m(2N^2+3N)((S+1)B)^{J+1}\delta.$$

Since $J = \lceil\frac{md-1}{S-1}\rceil \leq md - 1$, for the output function, we have

$$\left\|c \cdot h^{(J+1)} - \hat{c} \cdot \hat{h}^{(J+1)}\right\|_\infty \leq 150m^3d^2N^2((S+1)B)^{md}\delta := \hat{\delta}.$$

Then, by taking an $\delta$-net for each of $w^{(j)}$, $b^{(j)}$, $c$ and $F^{(J+1)}$, we know that the covering number of the hypothesis space $\mathcal{H}$ with radius $\hat{\delta} \in (0,1]$ can be bounded as

$$\mathcal{N}\left(\hat{\delta}, \mathcal{H}\right) \leq \left\lceil\frac{2B}{\delta}\right\rceil^{(S+1)J} \Pi_{j=1}^J \left\lceil\frac{(2(S+1)B)^j}{\delta}\right\rceil^{2S+1} \left\lceil\frac{(2(S+1)B)^{J+1}}{\delta}\right\rceil^{m(2N+3)}$$
$$\left\lceil\frac{2NB}{\delta}\right\rceil^{m(2N+3)} \left\lceil\frac{2}{\delta}\right\rceil^{m(2N+3)}$$
$$\leq \left(\frac{1}{\delta}\right)^{(S+1)J+J(2S+1)+3m(2N+3)} N^{m(2N+3)}(3(S+1)B)^{(S+1)J+\frac{J^2+J}{2}(2S+1)+m(2N+3)(J+3)}$$
$$\leq \left(\frac{1}{\hat{\delta}}\right)^{C_{S,d,m}N} N^{C'_{S,d,m}N}(153md(S+1)B)^{3mdC_{S,d,m}N},$$

where $C_{S,d,m} = 5(m^2d + 2m) + (md - 1)(mds + md + s + 1)$, $C'_{S,d,m} = 2C_{S,d,m} + 5m$. Thus we have

$$\log\left\{\mathcal{N}\left(\hat{\delta}, \mathcal{H}\right)\right\} \leq C_{S,d,m,B}N\log\left\{\frac{1}{\hat{\delta}}\right\} + C''_{S,d,m,B}N\log\{N\},$$

where $C''_{S,d,m,B} = 3mdC_{S,d,m}\log(153md(S+1)B) + C'_{S,d,m}$. $\qquad\square$

**Proof of Theorem 3.7**

*Proof of Theorem 3.7.* Since $f_{D,\mathcal{H}}$ is the empirical minimizer from (2.1), we have $\mathcal{E}_D(f_{D,\mathcal{H}}) \leq \mathcal{E}_D(h)$ for any $h \in \mathcal{H}$ and $\mathcal{E}_D(\pi_M f_{D,\mathcal{H}}) \leq \mathcal{E}_D(f_{D,\mathcal{H}})$. Then we can derive

$$\mathcal{E}(\pi_M f_{D,\mathcal{H}}) - \mathcal{E}(f_\rho) = \mathcal{E}(\pi_M f_{D,\mathcal{H}}) - \mathcal{E}_D(\pi_M f_{D,\mathcal{H}})$$
$$+ \mathcal{E}_D(\pi_M f_{D,\mathcal{H}}) - \mathcal{E}_D(h)$$
$$+ \mathcal{E}_D(h) - \mathcal{E}(h) + \mathcal{E}(h) - \mathcal{E}(f_\rho)$$
$$\leq \mathcal{E}(\pi_M f_{D,\mathcal{H}}) - \mathcal{E}_D(\pi_M f_{D,\mathcal{H}})$$
$$+ \mathcal{E}_D(h) - \mathcal{E}(h) + \mathcal{E}(h) - \mathcal{E}(f_\rho).$$

For simplicity, we denote

$$\mathcal{D}\left(\mathcal{H}\right) = \mathcal{E}\left(h\right) - \mathcal{E}\left(f_\rho\right) = \|h - f_\rho\|_{\rho_\mathcal{X}}^2\,,$$

$$S_1\left(n, \mathcal{H}\right) = \{\mathcal{E}_D\left(h\right) - \mathcal{E}_D\left(f_\rho\right)\} - \{\mathcal{E}\left(h\right) - \mathcal{E}\left(f_\rho\right)\}\,,$$

and

$$S_2\left(n, \mathcal{H}\right) = \{\mathcal{E}\left(\pi_M f_{D,\mathcal{H}}\right) - \mathcal{E}\left(f_\rho\right)\} - \{\mathcal{E}_D\left(\pi_M f_{D,\mathcal{H}}\right) - \mathcal{E}_D\left(f_\rho\right)\}\,.$$

Thus we have

$$\mathcal{E}\left(\pi_M f_{D,\mathcal{H}}\right) - \mathcal{E}\left(f_\rho\right) \le \mathcal{D}\left(\mathcal{H}\right) + S_1\left(n, \mathcal{H}\right) + S_2\left(n, \mathcal{H}\right).$$

Now we define the random variable $\eta$ on $\mathcal{Z}$ to be

$$\eta(z) = (y - h(x))^2 - (y - f_\rho(x))^2\,,$$

and $\sigma^2(\eta)$ is the variance. Then it can be easily derived that $|\eta(z)| \le (3M + \|h\|_\infty)^2$, $|\eta - \mathbb{E}\left(\eta\right)| \le 2\left(3M + \|h\|_\infty\right)^2$ and $\sigma^2(\eta) \le \mathbb{E}\left(\eta^2\right) \le \left(3M + \|h\|_\infty\right)^2 \mathcal{D}\left(\mathcal{H}\right)$. Then we can apply Lemma A.4 to get

$$\mathbb{P}\left\{S_1\left(n, \mathcal{H}\right) < \epsilon\right\} \ge 1 - \exp\left\{-\frac{n\epsilon^2}{2\left(3M + \|h\|_\infty\right)^2 \left(\mathcal{D}\left(\mathcal{H}\right) + \frac{2\epsilon}{3}\right)}\right\}.$$

Now we consider a function set

$$\mathcal{G} := \left\{\tilde{f} = \left(\pi_M f(x) - y\right)^2 - \left(f_\rho(x) - y\right)^2 : f \in \mathcal{H}\right\},$$

and for any fixed $\tilde{f} \in \mathcal{G}$, there exists an $f \in \mathcal{H}$ such that $\tilde{f}(z) = \left(\pi_M f(x) - y\right)^2 - \left(f_\rho(x) - y\right)^2$. Then we know that $\mathbb{E}\left(\tilde{f}\right) = \|\pi_M f - f_\rho\|_{\rho_\mathcal{X}}^2$ and $\frac{1}{n}\sum_{i=1}^n \tilde{f}\left(z_i\right) = \mathcal{E}_D\left(\pi_M f\right) - \mathcal{E}_D\left(f_\rho\right)$. It can be easily derived that $|\tilde{f}(z)| \le 8M^2$, $|\tilde{f}(z) - \mathbb{E}\left(\tilde{f}\right)| \le 16M^2$ and $\mathbb{E}\left(\tilde{f}^2\right) \le 16M^2\mathbb{E}\left(\tilde{f}\right)$. Then we can apply Lemma A.5 to $\mathcal{G}$ with $B' = \tilde{c} = 16M^2$ to get

$$\sup_{f \in \mathcal{H}} \frac{\mathcal{E}\left(\pi_M f\right) - \mathcal{E}\left(f_\rho\right) - \left(\mathcal{E}_D\left(\pi_M f\right) - \mathcal{E}_D\left(f_\rho\right)\right)}{\sqrt{\mathcal{E}\left(\pi_M f\right) - \mathcal{E}\left(f_\rho\right) + \epsilon}} \le \sqrt{\epsilon}, \tag{13}$$

holds with probability at least $1 - \mathcal{N}\left(\epsilon, \mathcal{G}, L^\infty\left(\mathcal{X} \times \mathcal{Y}\right)\right)\exp\left\{-\frac{3n\epsilon}{512M^2}\right\}$. Since for any $f_1, f_2 \in \mathcal{H}$, we have

$$\left|\tilde{f}_1 - \tilde{f}_2\right| \le 4M\left|f_1(x) - f_2(x)\right|.$$

Thus, an $\frac{\epsilon}{4M}$ covering of $\mathcal{H}$ provides an $\epsilon$ covering of $\mathcal{G}$ for any $\epsilon > 0$ which implies that

$$\mathcal{N}\left(\epsilon, \mathcal{G}, L^\infty\left(\mathcal{X} \times \mathcal{Y}\right)\right) \le \mathcal{N}\left(\frac{\epsilon}{4M}, \mathcal{H}, L^\infty\left(\mathcal{X}\right)\right).$$

Combining this with (9), we have

$$\mathcal{N}\left(\epsilon, \mathcal{G}, L^\infty\left(\mathcal{X} \times \mathcal{Y}\right)\right) \le \exp\left\{C_1 n_1 \log \frac{4M}{\epsilon} + C_2 n_2 \log n_2\right\}.$$

This together with (13) implies that

$$S_2\left(n, \mathcal{H}\right) \le \frac{1}{2}\left(\mathcal{E}\left(\pi_M f_{D,\mathcal{H}}\right) - \mathcal{E}\left(f_\rho\right)\right) + \epsilon,$$

holds with confidence at least $1 - \exp\left\{C_1 n_1 \log \frac{4M}{\epsilon} + C_2 n_2 \log n_2 - \frac{3n\epsilon}{128M^2}\right\}$. Combing all these inequalities, we have

$$\mathcal{E}\left(\pi_M f_{D,\mathcal{H}}\right) - \mathcal{E}\left(f_\rho\right) \le 2\mathcal{D}\left(\mathcal{H}\right) + 4\epsilon,$$

holds with probability at least $1 - \exp\left\{C_1 n_1 \log \frac{4M}{\epsilon} + C_2 n_2 \log n_2 - \frac{3n\epsilon}{128M^2}\right\} - \exp\left\{-\frac{n\epsilon^2}{2\left(3M + \|h\|_\infty\right)^2\left(\mathcal{D}(\mathcal{H}) + \frac{2\epsilon}{3}\right)}\right\}$. We finish the proof by letting $\delta = 4\epsilon$. $\qquad\square$

**Proof of Theorem 3.10**

*Proof of Theorem 3.10.* First, we associate a probability measure $\rho_f \in \mathcal{M}(\rho, \Theta)$ to a pair $(\mu, f)$ where $\mu$ is a measure on $\mathcal{X}$ and $f \in \Theta$. We assume that $\mu$ is upper and lower bounded by constants $\tau_1$ and $\tau_2$. Now we define a probability measure $\rho_f$ by

$$d\rho_f(x, y) = \left[ \frac{T + f(x)}{2T} d\delta_T(y) + \frac{T - f(x)}{2T} d\delta_{-T}(y) \right] d\mu(x), \tag{14}$$

where $T = 4mG$ and $d\delta_T$ denotes the Dirac delta with unit mass at $T$. It can be verified that $\rho_f$ is a probability measure on $\mathcal{X} \times \mathcal{Y}$ with $\mu$ being the marginal distribution $\rho_X$ and $f$ the regression function. Moreover, $M \geq 4mG$ ensures $|y| \leq M$ almost surely. Hence for any $f \in \Theta$, $\rho_f \in \mathcal{M}(\rho, \Theta)$.

Now we would apply **Theorem 2.7** in Tsybakov (2008) to prove our conclusion. It states that if for some $\widetilde{N} \geq 1$ and $\kappa > 0$, $f_0, \cdots, f_{\widetilde{N}} \in \Theta$ are such that

1. $\|f_i - f_j\|_{L^2_{\rho_\mathcal{X}}}^2 \geq \kappa n^{-\frac{2\alpha}{2\alpha+1}}$ for all $0 \leq i < j \leq \widetilde{N}$,

2. $\frac{1}{\widetilde{N}} \sum_{j=1}^{\widetilde{N}} \mathrm{KL}\left( \rho_j^n \| \rho_0^n \right) \leq \frac{\log \widetilde{N}}{9}$,

then there exists a positive constant $c_{\kappa, \tau_1, \tau_2}$ such that

$$\inf_{\hat{f}_n} \sup_{\rho \in \mathcal{M}(\rho, \Theta)} \mathbb{E} \left\| \hat{f}_n(x) - f_\rho(x) \right\|_{L^2_{\rho_\mathcal{X}}}^2 \geq c_{\kappa, \tau_1, \tau_2} n^{-\frac{2\alpha}{2\alpha+1}}. \tag{15}$$

Now we construct a finite sequence $f_0, \cdots, f_{\hat{N}_n}$ in the space $\Theta$. First, we let function $K \in L^2(\mathbb{R}) \cap \mathrm{Lip}^\alpha(\mathbb{R})$ be supported on $[-\frac{1}{2}, \frac{1}{2}]$ with Lipschitz constant $\frac{1}{2}L$ and $\|K\|_\infty \leq G$. Clearly this function exists. We partition the set $[-1, 1]$ into $\hat{N}_n = \lfloor c_\tau n^{\frac{1}{2\alpha+1}} \rfloor$ interval $\{A_{n,k}\}_{k=1}^{\hat{N}_n}$ with equivalent length $\frac{2}{\hat{N}_n}$, centers $\{u_k\}_{k=1}^{\hat{N}_n}$ and $c_\tau = \frac{2304\tau_1}{15T^2} \|K\|_2^2 + 1$. Now we define function as

$$\psi_{u_k}(x) = \frac{1}{\hat{N}_n^\alpha} K\left( \frac{1}{2} \hat{N}_n(x - u_k) \right), \quad \text{for } k = 1, \cdots, \hat{N}_n.$$

It is clear that $\psi_{u_k}(x)$ are Lipschitz-$\alpha$ functions with Lipschitz constant $\frac{1}{2^{1+\alpha}} L$ for $0 < \alpha \leq 1$ for $k = 1, \cdots, \hat{N}_n$ and $\|\psi_{u_k}(x)\|_\infty \leq G$. From the definition above, we can also see that for $u_i \neq u_j$, $\psi_{u_i}(x)$ and $\psi_{u_j}(x)$ have different supports. Now we consider the set of all binary sequences of length $\hat{N}_n$,

$$\Omega = \left\{ \omega = \left( \omega_1, \cdots, \omega_{\hat{N}_n} \right), \omega_i \in \{0, 1\} \right\} = \{0, 1\}^{\hat{N}_n},$$

and define functions $\phi_\omega(x)$ as

$$\phi_\omega(x) = \sum_{k=1}^{\hat{N}_n} \omega_k \psi_{u_k}(x).$$

Now we are going to show that $\phi_\omega(x)$ is a Lipschitz-$\alpha$ function with Lipschitz constant $L$ and $\|\phi_\omega(x)\|_\infty \leq G$ for any $\omega$. The sup-norm can be check easily by noticing that this is a summation on different supports and $|\omega_k| \leq 1$. Now we are going to check the Lipchitz constant. If $x, y \in A_{n,i}$, then we have

$$|\phi_\omega(x) - \phi_\omega(y)| = |\psi_{u_i}(x) - \psi_{u_i}(y)| \leq L |x - y|^\alpha.$$

If $x \in A_{n,i}$ and $y \in A_{n,j}$ for $i \neq j$, and we let $\bar{x}$ and $\bar{y}$ be the boundary points of $A_{n,i}$ and $A_{n,j}$ between $x$ and $y$, then we have,

$$
\begin{aligned}
&|\phi_\omega(x) - \phi_\omega(y)| \\
&= |\omega_i \psi_{u_i}(x) - \omega_j \psi_{u_j}(y)| \\
&\leq |\psi_{u_i}(x)| + |\psi_{u_j}(y)| \\
&= |\psi_{u_i}(x) - \psi_{u_i}(\bar{x})| + |\psi_{u_j}(y) - \psi_{u_j}(\bar{y})| \\
&\leq 2^{-1-\alpha} L \left( |x - \bar{x}|^\alpha + |y - \bar{y}|^\alpha \right) \\
&= 2^{-\alpha} L \left( \frac{1}{2} |x - \bar{x}|^\alpha + \frac{1}{2} |y - \bar{y}|^\alpha \right) \\
&\leq L \left( \frac{|x - \bar{x}| + |y - \bar{y}|}{2} \right)^\alpha \\
&\leq L |x - y|^\alpha,
\end{aligned}
$$

where we have applied Jensen's inequality. Then $\phi_\omega(x)$ is a Lipschitz-$\alpha$ function with Lipschitz constant $L$. Since for any $f \in \Theta$, it can be written in the form of $\sum_{i=1}^m g_i(\xi_i \cdot x)$ with $g_i \in W_\infty^\alpha[-1,1]$. Now we can simply take $m = 1$, $\xi_1 = [1, 0, \cdots, 0]$ and $g_1(x) = \phi_\omega(x_1)$. Then we denote $f_\omega(x) = \phi_\omega(x_1)$ where $x_1$ is the first component of $x \in \mathbb{R}^d$ and clearly we know that $f_\omega \in \Theta$.

For any $u_k$, we have $\|\psi_{u_k}\|_2^2 = \frac{2}{\hat{N}_n^{1+2\alpha}} \|K\|_2^2$. We use $\mathrm{Ham}(\omega, \omega') = \sum_{k=1}^{\hat{N}_n} I(\omega_k \neq \omega_k')$ to denote the Hamming distance between the binary sequences $\omega$ and $\omega'$. We know that

$$
\|f_\omega - f_{\omega'}\|_2^2 = \mathrm{Ham}(\omega, \omega') \frac{2}{\hat{N}_n^{1+2\alpha}} \|K\|_2^2.
$$

By the Varshamov-Gilbert bound (Lemma 2.9 in Tsybakov (2008)), we conclude that there exists a subset $\mathcal{W} \subset \Omega$ of cardinality $|\mathcal{W}| \geq 2^{\frac{\hat{N}_n}{8}}$ such that $\mathrm{Ham}(\omega, \omega') \geq \frac{\hat{N}_n}{8}$ for all $\omega, \omega' \in \mathcal{W}$, $\omega \neq \omega'$. Then we have

$$
\|f_\omega - f_{\omega'}\|_2^2 \geq \frac{1}{4} \hat{N}_n^{-2\alpha} \|K\|_2^2, \quad \text{for } \omega, \omega' \in \mathcal{W}, \omega \neq \omega'.
$$

Further, we know that

$$
\|f_\omega - f_{\omega'}\|_{L_{\rho_\mathcal{X}}^2}^2 \geq \kappa n^{-\frac{2\alpha}{2\alpha+1}}, \quad \text{for } \omega, \omega' \in \mathcal{W}, \omega \neq \omega',
$$

by taking $\kappa = \frac{1}{4} \tau_2 c_\tau^2 \|K\|_2^2$. The above inequality verifies the condition 1 in **Theorem 2.7** in Tsybakov (2008).

For simplicity, we use $f_i$ to denote $f_{\omega^i}$. Now we consider the KL-divergence $\mathrm{KL}\left(\rho_{f_i} | \rho_{f_j}\right)$. By the euation (14), we have $d\rho_{f_i}(x, y) = g(x, y) d\rho_{f_j}(x, y)$ with

$$
g(x, y) = \frac{T + \mathrm{sign}(y) f_i(x)}{T + \mathrm{sign}(y) f_j(x)} = 1 + \frac{\mathrm{sign}(y)(f_i - f_j)}{T + \mathrm{sign}(y) f_j}.
$$

Then we know that

$$
\begin{aligned}
&\mathrm{KL}\left(\rho_{f_i} | \rho_{f_j}\right) \\
&= \int_\mathcal{X} \frac{T + f_i(x)}{2T} \ln \left( 1 + \frac{f_i(x) - f_j(x)}{T + f_j(x)} \right) + \\
&\quad \frac{T - f_i(x)}{2T} \ln \left( 1 + \frac{f_i(x) - f_j(x)}{T - f_j(x)} \right) d\mu(x) \\
&\leq \int_\mathcal{X} \frac{f_i(x) - f_j(x)}{2T} \left( \frac{T + f_i(x)}{T + f_j(x)} - \frac{T - f_i(x)}{T - f_j(x)} \right) d_\mu(x) \\
&\leq \frac{16}{15T^2} \|f_i - f_j\|_{L_{\rho_\mathcal{X}}^2}^2.
\end{aligned}
$$

Since $\mathrm{KL}\left(\rho_{f_i}^n | \rho_{f_j}^n\right) \leq \frac{16}{15T^2} n \|f_i - f_j\|_{L_{\rho_\mathcal{X}}^2}^2$, we have

$$\mathrm{KL}\left(\rho^n_{f_i}|\rho^n_{f_j}\right) \leq \frac{16\tau_1}{15T^2} n \|f_i - f_j\|_2^2$$

$$\leq \frac{16\tau_1}{15T^2} n \mathrm{Ham}(\omega, \omega') \frac{2}{\hat{N}_n^{1+2\alpha}} \|K\|_2^2$$

$$\leq \frac{2304\tau_1}{15T^2} \|K\|_2^2 \frac{n}{\hat{N}_n^{1+2\alpha}} \frac{\hat{N}_n}{72}.$$

By $c_\tau = \frac{2304\tau_1}{15T^2} \|K\|_2^2 + 1$, we know that

$$\mathrm{KL}\left(\rho^n_{f_i}|\rho^n_{f_j}\right) \leq \frac{\hat{N}_n}{72} \leq \frac{\log |\mathcal{W}|}{9}.$$

Then two conditions of inequality (15) are satisfied by taking $|\mathcal{W}| = \tilde{N}$. Then we have

$$\inf_{\hat{f}_n} \sup_{\rho \in \mathcal{M}(\rho, \Theta)} \mathbb{E} \left\| \hat{f}_n(x) - f_\rho(x) \right\|_{L^2_{\rho_\mathcal{X}}}^2 \geq c_{\kappa, \tau_1, \tau_2} n^{-\frac{2\alpha}{2\alpha+1}}.$$

The proof can be applied directly to Corollary 3.11 by noticing that $x_1$ is a polynomial of input $x$. For simplicity, $\|K\|_2$ can be chosen to be 1. We know that $\tau_1$ and $\tau_2$ are actually upper and lower bounds of marginal density of $x_1$, the first component of $x$, thus we can simply take $\tau_1 = 1$ and $\tau_2 = \frac{1}{100}$. Since $\kappa$ is a constant depending on $\tau_1$, $\tau_2$, $m$ and $G$, the constant $c_{\kappa, \tau_1, \tau_2}$ essentially depends on $m$ and $G$.

This finishes the proof.

## Auxiliary Lemmas

**Lemma A.1.** *[Fang et al. (2020)] Given an integer $N$, let $\boldsymbol{t} = \{t_i\}_{i=1}^{2N+3}$ be the uniform mesh on $\left[-1 - \frac{1}{N}, 1 + \frac{1}{N}\right]$ with $t_i = -1 + \frac{i-2}{N}$. Construct a linear operator $L_{\boldsymbol{t}}$ on $C[-1, 1]$ by*

$$L_{\boldsymbol{t}}(f)(u) = \sum_{i=2}^{2N+2} f(t_i)\delta_i(u), \quad u \in [-1, 1], \ f \in C[-1, 1],$$

*where $\delta_i \in C(\mathbb{R})$, $i = 2, \ldots, 2N + 2$, is given by*

$$\delta_i(u) = N(\sigma(u - t_{i-1}) - 2\sigma(u - t_i) + \sigma(u - t_{i+1})). \tag{16}$$

*Then for $g \in C[-1, 1]$, $\|L_{\boldsymbol{t}}(g)\|_{C[-1,1]} \leq \|g\|_{C[-1,1]}$ and*

$$\|L_{\boldsymbol{t}}(g) - g\|_{C[-1,1]} \leq 2\omega(g, 1/N),$$

*where $\omega(g, \mu)$ is the modulus of continuity of $g$ given by*

$$\omega(g, \mu) = \sup_{|t| \leq \mu} \{|g(v) - g(v + t)| : v, v + t \in [-1, 1]|\}.$$

For the convenience of counting free parameter numbers, we introduce a linear operator $\mathcal{L}_N : \mathbb{R}^{2N+1} \to \mathbb{R}^{2N+3}$ given for $\zeta = (\zeta_i)_{i=1}^{2N+1} \in \mathbb{R}^{2N+1}$ by

$$(\mathcal{L}_N(\zeta))_i = \begin{cases} \zeta_2, & \text{for } i = 1, \\ \zeta_3 - 2\zeta_2, & \text{for } i = 2, \\ \zeta_{i-1} - 2\zeta_i + \zeta_{i+1}, & \text{for } 3 \leq i \leq 2N + 1, \\ \zeta_{2N+1} - 2\zeta_{2N+2}, & \text{for } i = 2N + 2, \\ \zeta_{2N+2}, & \text{for } i = 2N + 3. \end{cases} \tag{17}$$

An important property of the operator $\mathcal{L}_N$ is to express the approximation operator $L_{\mathbf{t}}$ on $C[-1,1]$ in terms of $\{\sigma\left(\cdot - t_j\right)\}_{j=1}^{2N+3}$ as

$$L_{\mathbf{t}}(f) = N \sum_{i=1}^{2N+3} \left(\mathcal{L}_N\left(\{f(t_k)\}_{k=2}^{2N+2}\right)\right)_i \sigma\left(\cdot - t_i\right), \quad \forall f \in C[-1,1]. \tag{18}$$

**Lemma A.2.** *[Zhou (2020b)] Let $S \geq 2$ and $W = (W_k)_{k=-\infty}^{\infty}$ be a sequence supported in $\{0,\cdots,\mathcal{M}\}$ with $\mathcal{M} \geq 0$. Then there exists a finite sequence of filters $\{w^{(j)}\}_{j=1}^{p}$ each supported in $\{0,\cdots,S\}$ with $p \leq \lceil \frac{\mathcal{M}}{S-1} \rceil$ such that the following convolutional factorization holds true*

$$W = w^{(p)} * w^{(p-1)} * \cdots * w^{(2)} * w^{(1)}.$$

**Lemma A.3.** *[Zhou (2020b)] Let $\{w^{(k)}\}_{k=1}^{J}$ be a set of sequences supported in $\{0,1,\ldots,S\}$. Then*

$$T^{(J)}\cdots T^{(2)}T^{(1)} = T^{(J,1)} := (W_{i-k})_{i=1,\ldots,d+JS,k=1,\ldots,d} \in \mathbb{R}^{(d+JS)\times d}, \tag{19}$$

*is a Toeplitz matrix associated with the filter $W = w^{(J)} * \cdots * w^{(2)} * w^{(1)}$ supported in $\{0,1,\cdots,JS\}$.*

**Lemma A.4.** *[Cucker & Zhou (2007)] Let $\eta$ be a random variable on a probability space $\mathcal{Z}$ with mean $\mathbb{E}(\eta) = \mu$, variance $\sigma^2(\eta) = \sigma^2$, and satisfying $|\eta(z) - \mathbb{E}(\eta)| \leq B_\eta$ for almost $z \in \mathcal{Z}$. Then for any $\epsilon > 0$,*

$$\mathbb{P}\left\{\frac{1}{m}\sum_{i=1}^{m}\eta(z_i) - \mu < \epsilon\right\} \geq 1 - \exp\left\{-\frac{m\epsilon^2}{2\left(\sigma^2 + \frac{1}{3}B_\eta\epsilon\right)}\right\}.$$

**Lemma A.5.** *[Zhou & Jetter (2006)] Let $\mathcal{G}$ be a set of continuous functions on $\mathcal{Z}$ such that for some $B' > 0$, $\tilde{c} > 0$, $|\tilde{f} - \mathbb{E}(\tilde{f})| \leq B'$ almost surely and $\mathbb{E}(\tilde{f}^2) \leq \tilde{c}\mathbb{E}(\tilde{f})$ for all $\tilde{f} \in \mathcal{G}$. Then for any $\epsilon > 0$,*

$$\mathbb{P}\left\{\sup_{\tilde{f}\in\mathcal{G}}\frac{\mathbb{E}(\tilde{f}) - \frac{1}{m}\sum_{i=1}^{m}\tilde{f}(z_i)}{\sqrt{\mathbb{E}(\tilde{f}) + \epsilon}} > \sqrt{\epsilon}\right\} \leq \mathcal{N}(\epsilon,\mathcal{G},L^\infty(\mathcal{X}))\exp\left\{-\frac{m\epsilon}{2\tilde{c} + \frac{2B'}{3}}\right\}.$$

$\square$

