# OpenReview forum: "Optimal Convergence Rates of Deep Convolutional Neural Networks: Additive Ridge Functions"
_TMLR — Accepted by TMLR_

### Review · Reviewer_ckgW · 2022-11-20

**Summary Of Contributions:**

This paper studies the learning ability of deep convolutional neural networks (DCNNs). In more details, the paper considers hypothesis of the form $c\cdot h^{(J+1)}(x)$, where $c$ is a weight vector and $h^{(J+1)}(x)$ is the output of a DCNN followed by a fully-connected layer. The paper shows that by putting appropriate constraints on the vector $c$ and the weights in DCNNs, the functions of the form $c\cdot h^{(J+1)}(x)$ can achieve minimax optimal rates (up to log factors) for learning additive ridge functions. This result is derived by computing the covering numbers of the spaces of DCNNs and concentration inequalities to derive estimation error bounds. These estimation error bounds are further combined with the existing approximation error bounds. Matching lower bounds are also presented.

**Audience:**

Yes

**Broader Impact Concerns:**

There are no broader impact concerns since this submission is of theoretical nature.

**Claims And Evidence:**

Yes

**Requested Changes:**

I would like the authors to address the issues on the computation of covering numbers. I would also request the changes on the typos.

**Strengths And Weaknesses:**

**Strength**

The paper considers an interesting problem and is very well written. The developed covering number bounds and excess risk bounds are very interesting to understand the behavior of DCNNs. In particular, the derived bounds are dimension-independent with the input dimension appearing only in the constant.The proofs are easy to follow and seem to be correct except minor issues which can be easily addressed.

**Weakness**

In the proof of Lemma 3.5 on covering numbers, the authors show

\begin{align*}
N(\hat{\delta}, H) &\leq c(1/\delta)^{(S+1)J}\prod_{j=1}^J(1/\delta)^{2S+1}(1/\delta)^{m(2N+3)}(1/\delta)^{m(2N+3)}(1/\delta)^{m(2N+3)}\\\\
& \leq (1/\delta)^{(S+1)J+2^{-1}(J^2+J)(2S+1)+m(2N+3)(J+3)},
\end{align*}
where $c$ is independent of $\delta$. However, according to the first inequality, the second inequality should be
$$
N(\hat{\delta}, H) \leq  (1/\delta)^{(S+1)J+(2S+1)J+3m(2N+3)}.
$$
I would suggest the authors check this computation.

In the definition of covering number, it has two arguments. However, in the proof of Thm 3.6, it has three arguments.

**Minor comments**

- Lemma 3.5: it seems that $C''_\{S,dm\}$ should be $C''_\{S,d,m,B\}$

- Section 3.4: "is drawn from and"

- Below eq (15): $|||K||_\infty$ should be $||K||_\infty$

- page 18: "is a constant depends on" should be "is a constant depending on"

- Eq (19): the last period should be removed

- Proof of Thm 3.9: "is an measure" should be "is a measure"

- In the proof of Thm 3.6: the meaning of $\sigma^2$ is not given.

- In the proof of Thm 3.6: the authors use the notation $E[\xi]$. In Thm 3.7, the authors use the notation $\mathbb{E}$. The authors should use a consistent notation for the expectation operator.

---

> ### Author Response · Authors · 2022-11-30
> **For Reviewer ckgW**
>
> We thank the reviewer for the encouraging comments and suggestions. We appreciate the time you spent on the paper. Below we address the concerns and comments that you have provided.
>
> **Q**: *In the proof of Lemma 3.5 on covering numbers*...
>
> **A**:Thank you very much for your careful review. We checked the proof of Lemma 3.5 and we did make this mistake here. We will fix this problem in the next version.
>
> **Q**: *In the definition of covering number, it has two arguments. However, in the proof of Thm 3.6, it has three arguments.*
>
> **A**: The third argument represents the metric we consider. We will clarify this in the definition.
>
> **Q**: *Minor comments*
>
> **A**: Thank you very much for your careful reading, we will fix these in the next version.

---

### Review · Reviewer_aznz · 2022-11-23

**Summary Of Contributions:**

This paper studies the regression properties of a certain form of convolutional neural networks where the target function is an additive ridge function. It provides new theoretical results on the mean squared error. The contributions are as follow:

1. Leveraging existing approximation results in the case of a regression problem with additive ridge target function, the authors obtain an upperbound on the mean squared error by carefully bounding the covering number of the hypothesis class.

2. A mini-max lower rate is also obtained with this framework.


**Audience:**

Yes

**Claims And Evidence:**

Yes

**Requested Changes:**

Overall, there is nothing to criticise about the theoretical correctness of this paper and I think it fits in the scope of TMLR. However, I would like to see a comprehensive discussion/answer to my questions above, as I believe it would clarify better the contributions.

**Other minor changes**: there are multiple claims that I found either inaccurate or wrong:

- In the introduction, the authors claim that `convolutional neural networks are considered as the state-of-the-art  for various tasks such image classification'. While this was true for a long time, it is no longer the case as now Vision Transformers beat ConvNets in many (almost all) standard image tasks.

- In the problem setting section, the authors claim that "For convolutional networks, the number of parameters can be decreased dramatically" because of the weight-sharing property. This is not true in general since in practice, convolutional layers are defined with multiple filters, and with a large number of filters, the convnet can exceed the fully-connected network in the number of parameters. I would not use this as a selling point for the theoretical analysis of ConvNets.

- Other minor issues: I would not use the expression "extraordinary abilities", something like "impressive abilities" is better. Same for "delicate covering...", I suggest "in-depth analysis ...". "The target of mean squared" $\to$ "The aim/goal of mean squared".

**Strengths And Weaknesses:**

### Strengths

The authors combine different ingredients to derive the results.
1. The covering number is derived using standard tools on this topic. A concentration-type inequality is then proven (Thm3.6) which will be a key factor in the main results.
2.  The authors then opportunely use exiting results on the approximation rates in the case of an additive ridge function and well-defined hypothesis class that consists of stacked convolutional layers followed by a fully-connected layer.
3. Using standard regression analysis, the authors derive an upper bound on the mean squared error (Thm3.7) and a mini-max lower rate (Thm3.9).
4. In addition to these technical contributions, the paper is well written and easy to follow. The proofs are concise and rigorous.

### Weaknesses

I have a few questions regarding the main results.

1. **Is it really a CNN analysis?** The hypothesis class consists of neural networks constructed with multiple convolutional layers followed with a fully-connected layer. I have read the proofs in Fang et al. (2020) and it seems that the only time the convolutional structure was used is to approximate a linear function of type $\xi \cdot x$ (see e.g. Lemma 5 in Fang et al. (2020)). This works in this case because the target function has the special additive ridge form which. Hence, is this result really leveraging the Convolutional structure? Another thing that got my attention is the dependence of the approximation rate in Thm3.1 (Thm2 in Fang et al.) on the number of parameters in the model. Achieving zero approximation rate requires only $N$ to go to infinity, which is exaclty the number of parameters in $c$ (a.k.a the output layer, right after the fully-connected one). The convolutional layers can have a fixed number of parameters and we still achieve a perfect match. As such, it seems that the role of convolutions here is just to replicate the linear operations of type $\xi \cdot x$ which is readily available if we just use a linear layer.

2. **What happens if we were to replace CNN with Fully-Connected?** could the authors provide some insights on how things will change if we consider a simple hypothesis class consisting of a single hidden layer fully connected neural networks? I assume the same approximation rates (if not better) should be achieved when the target function is a ridge additive function.

3. **What happens with multiple filters?** The hypothesis class used to obtain the results consists of convolutional layers with single filters. In practice, a convolutional layer usually have multiple filters, and the output of the layer is just a sum over the convolution operations with the different filters. Could you provide some insights on how the results would change if the hypothesis class incorporates multi-filters convolutional layers?


4. **What happens with padding?** The hypothesis class consists of a bit weird convolutional networks. The network width grows with depth, and the network is kind of triangular. This is because no padding is considered to preserve the dimensionality of the layers. What happens if, say, we consider circular padding? or just trim the first $S$ rows and the last $S$ rows from the matrix $T^w$?

---

> ### Author Response · Authors · 2022-11-30
> **For Reviewer aznz**
>
> We thank the reviewer for the encouraging comments and suggestions. We appreciate the time you spent on the paper. Below we address the concerns and comments that you have provided.
>
> **Q**: *Is it really a CNN analysis?*
>
> **A**: Your understanding of our strategy is correct. We first learn inner products $\xi_i \cdot x$ through the convolutional layers, then learn the function $f$ through the final layer. The analysis of convolutional neural networks is indeed relatively difficult, especially to prove the superiority of convolutional neural networks over fully connected networks. Because from a high-level point of view, any convolutional neural network is essentially just a sparse fully connected network with a special structure. Therefore, it is very natural to find an equivalent fully connected neural network in the end by analyzing directly from the convolutional neural network. As you pointed out in the review, the basis of our theoretical analysis comes from Lemma 5 in Fang et al. (2020), which proves that a large convolution can be split into several small convolutions. And this just shows that we are analyzing from the nature of convolution, so I believe that the analysis in this paper belongs to the category of convolutional neural network.
>
> **Q**: *What happens if we were to replace CNN with Fully-Connected?*
>
> **A**: Yes. The same approximation rate can be achieved for this type of function by a two layer fully connected networks with the same order of free parameters . For the first layer, by setting $b_i$ large enough, we have $\sigma(\xi_i \cdot x + b_i) = \xi_i \cdot x + b_i$. For the second layer, with a suitable construction, we can obtain the result in equation (12). Then we can apply Lemma A.1. to get the final approximator.
>
> **Q**: *What happens with multiple filters?*
>
> **A**:  If we consider this problem from the perspective of Lemma 5 in Fang et al. (2020) and fix or increase the number of channels, we can obtain several $W_i * x$ after convolutional layers where each $W_i$ is a large filter. Then all the information in vectors $W_i * x$ can be used for further construction. In this paper, $W$ is the concatenation of $m$ $\xi_i$ and we only use $m$ values out of the vector $W * x$. If we use multiple convolutional filters, then the target function space can be more complicated. As a simple extension of the target function in this paper, we can take the target function in the form of $\sum_{i=1}^m f_i( \sum_{j=1}^n \xi_{i,j} \cdot x)$ if we consider $n$ filters in each layer.
>
> **Q**: *What happens with padding?*
>
> **A**: I believe that with our current technical method, we can not consider the situation of circular padding or just simplify the matrix $T^w$. The main difficulty also lies in Lemma 5 in Fang et al. (2020) which is first proved in Section 5 of the paper "Deep distributed convolutional neural networks: Universality, Ding-Xuan Zhou (2018)". In the proof, the symbol $\tilde w$ of a filter mask $w$ supported on $\{0,1,\cdots,\tau\}$ is introduced and $\tilde w(z) = \sum_{k=0}^{\tau}w_kz^k$ is a polynomial on $\mathbb{C}$. A key property $\tilde{a*b}(z) = \tilde{a}(z)\tilde{b}(z)$ is applied where * stands for an ordinary convolution. If we consider circular padding, this will change the vector after each convolution (adding extra information that ordinary convolution does not contain) and the conclusion will no longer hold. The same situation will happen if we simply the matrix $T^w$ since we will lose some information in the convolution.
>
> **Q**: *Other minor changes*
>
> **A**: Thanks for your comments, we will fix these issues in the next version.

---

> > ### Comment · Reviewer_aznz · 2023-01-06
> > **Response**
> >
> > Sorry for the late reply. Could you please highlight the changes you made in the revised version (maybe in a different color)? I do not see any discussion on the limitations of this approach in the revised version. I believe a comprehensive discussion of the limitations (as discussed in my review above) will shed more light onthe contributions of this paper.

---

> > > ### Author Response · Authors · 2023-01-07
> > > **For Reviewer aznz**
> > >
> > > Dear Reviewer aznz:
> > >
> > > Sorry for the late reply. We have highlighted changes in the revised version with red color. Major limitations of our network structure have been stated in Remark 3.2.

---

> > > > ### Comment · Reviewer_aznz · 2023-01-09
> > > > **response**
> > > >
> > > > Thanks. Some minor issues :
> > > > * "a in-depth" --> "an in-depth"
> > > > * "We try to avoid this problem by considering convolutional neural networks" --> I suggest you remove this paragraph. Many standard CNNs have a large number of channels and this argument no longer holds.

---

> > > > > ### Author Response · Authors · 2023-01-09
> > > > > **For Reviewer aznz**
> > > > >
> > > > > Dear Reviewer aznz:
> > > > >
> > > > > Thank you very much for your reply. We have made changes according to your suggestions in the latest version of the paper.

---

### Review · Reviewer_eRfE · 2022-12-10

**Summary Of Contributions:**

This paper studies the learning ability of convolutional neural networks followed by one fully connected layer and shows a minimax rate for learning ridge functions using the minimizer of the empirical loss function. To achieve the result, the authors base the analysis on an existing approximation result, made careful estimate to the magnitude of the filtered, and derived a bound for the cover number.



**Audience:**

Yes

**Broader Impact Concerns:**

I have no concerns on the broader impact of this paper.

**Claims And Evidence:**

Yes

**Requested Changes:**

I have no required changes for the authors. It will be great if the authors can make some discussion on the two questions I listed above.

**Strengths And Weaknesses:**

I finds this result interesting. But I am also interested in more potential extensions to the result, specifically on the following aspects: (Of course, may not necessary for the current paper)

The training: this paper studies the properties of the minimizer of empirical loss. Surely this is standard practice in statistical learning theories. However, for neural networks it is hard to guarantee that the minimizer can be found. How do the authors comment on the optimization issue? Is it still possible to derive similar bounds when optimization is taken into consideration, even just for very simple networks?

The input shape: Convolutional neural networks are usually applied on two-dimensional arrays (images). Can the analysis be extended to image inputs?

---

> ### Author Response · Authors · 2022-12-19
> **For Reviewer eRfE**
>
> We thank the reviewer for the encouraging comments and suggestions. We appreciate the time you spent on the paper. Below we address the concerns and comments that you have provided.
>
> **Q**:*The training: this paper studies the properties of the minimizer of empirical loss. Surely this is standard practice in statistical learning theories. However, for neural networks it is hard to guarantee that the minimizer can be found. How do the authors comment on the optimization issue? Is it still possible to derive similar bounds when optimization is taken into consideration, even just for very simple networks?*
>
> **A**: Thank you very much for your question. In response to this question, we recommend the following two related papers to you.''Optimal Rates for Averaged Stochastic Gradient Descent under Neural Tangent Kernel Regime, Nitanda and Suzuki (2021)'' and "Regularization matters: A nonparametric perspective on overparametrized neural network, Hu Tianyang et al. (2021)". Both papers combine the optimization problem with statistical learning theory and derive specific orders of convergence rates. These two papers considered a one hidden layer neural network with infinite width which fit in the Neural Tangent Kernel (NTK) regime. The NTK (Jacot et al., 2018) has provided a connection between the learning process of a neural network and a kernel method in a reproducing kernel Hilbert space associated with an NTK. The global convergence of the gradient descent was demonstrated in (Du et al., 2019), (Allen-Zhu et al., 2019) through the development of a theory of NTK with the over-parameterization. In these theories, the positivity of the Gram-marix of the NTK leads to the equivalence between two learning dynamics for neural networks and kenrel methods with the NTK through a linear approximation of neural networks. In this setting, the training error can be well controlled, so it can be combined with statistical learning theory. But it is precisely because of such a special network structure that it is difficult to extend such theoretical results to general network structures.
>
> **Q**: *The input shape: Convolutional neural networks are usually applied on two-dimensional arrays (images). Can the analysis be extended to image inputs?*
>
> **A**: The basis of our theoretical analysis comes from Lemma 5 in Fang et al. (2020), which proves that a large convolution can be split into several small convolutions. However, this lemma only fit in the case of one-dimensional convolution. Based on the theoretical results of this paper alone, it is difficult to directly extend to the case of two-dimensional convolution. There have been some works considering two-dimensional convolutional structures, for example, "Approximation properties of deep ReLU CNNs, He et al., (2022)" and "On the rate of convergence of image classifiers based on convolutional neural networks, Kohler et al.,(2022)". However, there is currently no paper that directly approximates a specific function class (such as Sobolev space) through a two-dimensional convolutional neural network which is critical for the analysis of statistical learning theory. We are very much looking forward to getting satisfactory theoretical results in this direction.

---

### Decision · Action_Editors · 2023-01-15

**Recommendation:** Accept as is

**Comment:**

During the discussion period, the questions and concerns raised by the reviewers are well addressed by the authors. Overall, the reviewers are all positive about the technical contribution and the results obtained.   All the reviewers have recommended its acceptance.

**Audience:**

Puyu Wang, Quanguan Gu, Yuan Cao

**Claims And Evidence:**

This paper provided theoretical results on the mean squared error for certain convolutional neural networks with the assumption that the target function is an additive ridge function.  The main tools are the covering number and a newly developed concentration inequality.  The reviewers agree that the results are new and interesting. The paper is well-written and the proof is easy to follow.